# Towards a Distributed Digital Twin Framework for Predictive Maintenance in Industrial Internet of Things (IIoT)

**DOI:** 10.3390/s24082663

**Published:** 2024-04-22

**Authors:** Ibrahim Abdullahi, Stefano Longo, Mohammad Samie

**Affiliations:** School of Aerospace, Transport and Manufacturing (SATM), Cranfield University, Bedford MK43 0AL, UK; s.longo@cranfield.ac.uk (S.L.); m.samie@cranfield.ac.uk (M.S.)

**Keywords:** digital twins, predictive maintenance, wind turbines, fog computing, machine learning

## Abstract

This study uses a wind turbine case study as a subdomain of Industrial Internet of Things (IIoT) to showcase an architecture for implementing a distributed digital twin in which all important aspects of a predictive maintenance solution in a DT use a fog computing paradigm, and the typical predictive maintenance DT is improved to offer better asset utilization and management through real-time condition monitoring, predictive analytics, and health management of selected components of wind turbines in a wind farm. Digital twin (DT) is a technology that sits at the intersection of Internet of Things, Cloud Computing, and Software Engineering to provide a suitable tool for replicating physical objects in the digital space. This can facilitate the implementation of asset management in manufacturing systems through predictive maintenance solutions leveraged by machine learning (ML). With DTs, a solution architecture can easily use data and software to implement asset management solutions such as condition monitoring and predictive maintenance using acquired sensor data from physical objects and computing capabilities in the digital space. While DT offers a good solution, it is an emerging technology that could be improved with better standards, architectural framework, and implementation methodologies. Researchers in both academia and industry have showcased DT implementations with different levels of success. However, DTs remain limited in standards and architectures that offer efficient predictive maintenance solutions with real-time sensor data and intelligent DT capabilities. An appropriate feedback mechanism is also needed to improve asset management operations.

## 1. Introduction

From this Bloomberg news headline [1] and others alike, wind turbine failure is an expensive challenge facing the wind energy industry across the US and Europe. While research on predictive maintenance of wind turbines has gained traction, standardizing the application of technologies such as digital twins offer benefits that could better manage industrial assets. This research work builds on existing research on predictive maintenance by providing a distributed DT framework to improve predictive maintenance in manufacturing assets.

Since the emergence of Industry 4.0, digital twin (DT) appears to be one of the leading technologies towards digital transformation, especially in manufacturing 4.0. While there are several attempts to define DT, there is still no standardized and widely acceptable definition of it. A description of DT that is good enough to deter the usual misconceptions was described in [2], in which the authors differentiated a DT as having a two-way automatic data flow between the physical and digital object, rather than a manual or semi-automatic two-way data flow, which is more aligned with a digital model and a digital shadow. Clarifying misconceptions such as labeling DT as merely a simulation (i.e., a digital model) can inform researchers of the potential of using DTs beyond a limited application domain, but they must be utilized properly. This is why DT frameworks and standards such as the one in this work are relevant. The description of DT in [2] highlights the relevance of a two-way automated feedback necessary for utilizing “intelligent DTs”. An intelligent DT, in this context, can be described as a digital twin solution that monitors an asset in real time, predicts its future behavior, and reacts to potential issues by analyzing the best operational control mechanisms needed to handle the potential failure or at least reduce its impact. Researchers in [3,4] have shown that more than 85% of DTs in industrial sectors are for manufacturing assets, and [4] went further to highlight that from 2019 onwards, more than 90% of DT applications are in maintenance, followed by Prognostics & Health Management (PHM), and then other process optimizations. In terms of maintenance, predictive maintenance is the leading type of maintenance preferred by industries [4] because of its potential to save cost, time, and resources by anticipating downtimes and avoiding them way ahead of time. Our work in this paper revolves around developing a digital twin of an asset in operation rather than of lifecycle management.

A concise systematic literature review on predictive maintenance was performed in [5], where the authors answered key questions, having extensively carried out a literature review with aligned research questions on using DTs for predictive maintenance. With 10 research questions and 42 primary studies reviewed, this work identified a challenge of computational burden and lack of reference architectures in using DTs for predictive maintenance. This work builds on some of these gaps, challenges, and existing works to cater for the need for a digital twin framework that can improve predictive maintenance of an Industrial Internet of Things (IIoT) asset. The literature review section analyzes this in detail.

While wind turbines are generally considered part of the energy sector, and, specifically, the renewable energy sub-sector, some of its key components like the gearbox, generator, etc., are considered as products of the manufacturing sector. The maintenance of wind turbines, based on their complex engineering design, is expensive, as they can be onshore or offshore. This makes the Operation and Maintenance (O&M), as well as data gathering, connectivity, and remote monitoring tasks, intensive. Whether in a small wind farm with a few wind turbines or a large wind farm with a few thousand wind turbines that generate gigawatts of electricity, maintenance is a key aspect of the success of wind energy.

This paper explores the requirements of developing a predictive maintenance DT using a distributed architecture to address limitations of existing DT implementations found in the literature with regards to standards and reference architectures, computational latency, accuracy, and prediction feedback loop in real-time scenarios. The proposed framework aims to adopt a software engineering approach such as Object-Oriented Concepts and Software Development Life Cycle (SDLC) [6] and a distributed cloud computing paradigm-Fog Computing [7]. The ISO 23247 standard for digital twins in manufacturing [8] is used to guide the development of the framework. Overall, this framework contributes the benefits of improved real-time monitoring and accuracy of applying a distributed architecture to enhance the effectiveness of a PHM solution.

This work explored some essential papers in the literature review to identify the research gap. Exploring the use and application of DTs for predictive maintenance can be a broad area to cover. This is because many works in the literature have discussed the topic in part. In this work, we explore some relevant research outputs that have discussed some or related concepts. These research reviews focused on a “predictive maintenance digital twin” or implemented the technology associated with digital twin and predictive maintenance. The key question we attempt to answer is of how a distributed digital twin framework can improve the efficiency of predictive maintenance of a manufacturing asset. To break down these questions, we aim to answer the following.

How can digital twins support real-time predictive maintenance?What benefits will a digital twin framework implemented based on a standardized framework offer to a predictive maintenance solution in IIoT?How can this proposed digital twin framework be extended to act as an intelligent digital twin with a prediction feedback loop?

The objective of this paper is, therefore, to highlight a case study of implementing a predictive maintenance digital twin of wind turbines in a wind farm. The framework is aimed at showcasing the benefit of utilizing distributed DT nodes to enhance feedback (from digital twin to physical asset) while maintaining acceptable accuracy of the models. As DTs can also serve as simulation tools for “what-if-scenarios”, the distributed nodes aim to provide computational capacity to run and test on how reconfigurations to operational conditions can help manage or mitigate an imminent component failure.

The remainder of this paper is organized as follows. Section 2 discusses the literature review on DT and predictive maintenance along with DT architectures in the context of this work. Section 3 establishes the theoretical framework of our proposed system architectural framework. Section 4 explains the methodology of implementing the key technology aspects considered in the framework. Section 5 describes the experimental set up, dataset, and selection of key components of the wind turbines. Section 6 discusses the results, and Section 7 concludes the paper with a highlight of future work.

## 2. Literature Review

This section reviews the literature and the key technologies involved in answering the questions identified above.

### 2.1. Digital Twins and Predictive Maintenance

Digital Twins’ two-way automatic data flow [4] makes them suitable for predictive maintenance (PdM). Several authors adopted the DT term from its initial introduction by Michael Grieves [9], along with NASA’s broad definition and adoption of the term [10], seeing it as “an integrated multi-physics, multiscale, probabilistic simulation of an as-built vehicle or system that uses the best available physical models, sensor updates, fleet history, etc., to mirror the life of its corresponding flying twin”. However, like NASA’s adoption of DT, many authors derive a definition based on the specific use case their work covers. From the review of literature [11], a DT can be simply put as a replication of a physical object in the digital space, with a connection that links them with data synchronization and status updates. As mentioned earlier, in this work, we perceive DT as a copy of the physical asset that has access to its operating data and is hosted in a computational platform that can use the data for PdM (utilizing data, algorithm, and platform).

Work by [12] showed a model for predictive maintenance using Asset Administration Shell (AAS), which uses a modular to decouple components into information and functions, linking with a semantic repository for each sub-model. The authors in [12] used a centralized approach for machine learning-based predictive maintenance. In this paper, we utilize the Asset Administration Shell in the form of the Digital Twin Definition Language (DTDL), which is a Microsoft Azure open digital framework [13].

Predictive maintenance, on the other hand, was described in [4] as a prognosis that uses all the information surrounding a system to predict its remaining life or when it is likely to fail. Developing a predictive maintenance model can be a model-driven approach based on analytical, physical, or numerical models, or it can be a data-driven approach based on data obtained from sensors. Both PdM approaches have been explored highly in Manufacturing 4.0, and it has been agreed that they are computationally expensive [4]. The data-driven approach of predictive maintenance relies on Internet of Things (IoT) to gather sensor readings from assets to use them in the digital space (computational platform). This work leverages the Industrial Internet of Things (IIoT) concept and thus uses the data-driven approach to implement the proposed distributed digital twin PdM solution based on sensor data acquired from wind turbines. However, the digital twin framework presented in this paper here can serve as a basis for utilizing physics-based model-driven approaches by the provision of the system architecture with a computational platform capable of handling simulations.

The authors in [14] developed a predictive digital twin for offshore wind turbines in which they used the prophet algorithm as a time series prediction model. The DT in [14] was developed in Unity3D to have a visual sense of the operating conditions of the wind turbines using the OPC-Unified Architecture (OPC-UA) as the data communications protocol that streams live data to the DT. Ref. [14] used vibration and temperature data for their model and the choice of prophet model was to factor in seasonality. The root mean square (RMS) achieved showed they were able to predict failure before it occurred. The authors of [14] suggested the DT feedback was based on the ability of a user to forecast future failure from the DT. The implemented DT in this work, our framework, supports an “intelligent DT” that enhances two-way feedback.

Work by the authors of [15], implemented a PdM solution based on SCADA data of the generator and gearbox of wind turbines using three algorithms: XGBoost, Long Short-Term Memory (LSTM), and Multiple Linear Regression (MLR). The authors evaluated the algorithm performance using R-squared, RMSE, MAE, and MAPE and used Statistical Process Chart (SPC) to detect anomalous behavior. In the results of [15], the models predicted failure up to six days before its occurrence, with LSTM outperforming XGBoost for the generator and vice-versa for the gearbox. There was no DT or any feedback mechanism in this work [15]. Another work [16] applied a data-driven approach (decision trees) with a focus on the data pre-processing using hyper-parameter tuning to detect failures from five components of a wind turbine, mainly the generator, hydraulic, generator bearing, transformer, and gearbox. They showed how a good pre-processing strategy in data-driven models can outperform a model-driven approach for PdM.

The paper [17] introduced a cyber-physical CPS architecture for PdM with several modules for condition monitoring, data augmentation, and ML/DL, which supports an intelligent decision-making strategy for fault prediction with good KPI. The architecture [17] used both MQTT as a communications protocol and OPC/UA for industrial automations.

On the distributed digital twin concept, work by [18] presented the concept of “IoTwins”, in which the authors argue that the best strategy for implementing DTs is to deploy them close to the data sources to leverage IoT gateway on edge nodes and use the cloud for heavier computational task such as machine learning model training. In their reference architecture, the authors outline how the edge-fog-cloud paradigm can allow a distributed DT to leverage the needed layers of computing and interfaces that are suitable for real-time applications. A similar concept was introduced in our earlier work [19].

Considering reference architectures for DT, work by the authors of [20] used the ISO 23247 to develop a DT for additive manufacturing that resolved interoperability and integration issues in real-time decision making and control. Work by the authors of [20] utilized the ISO23247 to develop a data mapping approach called “EXPRESS Schema” that uses the edge for data modeling of a DT of a machine tool.

A further review in [21] of unit-level DT for manufacturing components explored the current stage of research into the application of DT at the unit level to foster real-time control. The output in [21] is an extensive analysis that clearly shows a gap in the need for adopting standards, handling hardware and software requirements, as well as a mechanism for DT to physical twin (PT) feedback. An abstract of [21] is shown in Figure 1. The highlighted gaps directly align with the output of this paper. 

Leading Cloud technology providers like IBM, Microsoft, and AWS have outlined DT reference architectures owing to the need of significant connectivity and computing power to run and manage DTs at scale [22]. 

Out of the reviewed works, it has been observed from the literature that few attempts have been made to showcase the benefits of distributed DT implementation for predictive maintenance as a manufacturing industry solution. As such, this work will go further to present a DT framework that attempts to cover these gaps [21,23], summarized in Figure 1.

### 2.2. Computing Infrastructure for Digital Twins

#### 2.2.1. Cloud, Fog, and Edge Computing

Cloud computing refers to the on-demand delivery of computing services over the internet [24,25,26], and these computing services that range from computing, storage, networking, and other tools make the cloud a suitable platform for the deployment of DTs.Fog computing is an extension of the cloud, introduced by Cisco in 2012 [7], as a concept that brings computing power closer to the data sources, thereby reducing latency and improving other computational benefits.Edge computing is like the fog computing concept. It deals with the ability for Internet of Things (IoT) devices distributed in remote locations to process data at the “edge” of the network [27].Industrial Internet of Things (IIoT) is simply the connectivity of objects to transmit data between other objects and the internet. The application of IoT to industrial applications such as in manufacturing, energy, etc., is termed Industrial IoT [28]. Industry 4.0 is the digital transformation that led to enhancement of manufacturing equipment “from steam to sensor” [29], thereby enabling them to connect, transmit, and process data in the cloud or locally. The DT framework in this work leverages the IIoT concept to create the digital twin through real-time streaming of sensor readings from the physical asset.

To highlight the difference with context, edge computing is usually distinctively recognized when this processing is performed by billions of IoT devices, and when dedicated local servers in millions are involved, it is termed fog computing [27]. Cisco [7] describes the fog-edge computing architecture as “decentralizing a computing infrastructure by extending the cloud through the placement of nodes strategically between the cloud and edge devices”.

As discussed in the distributed DT paper “IoTwins” [18], a DT implementation can utilize the edge-fog paradigm to leverage both hardware and software services to deploy DTs of manufacturing assets for seamless and efficient monitoring and application of data-driven ML solutions or model-driven simulation of physical assets. In this section, we primarily look at related attempts and technologies of the key aspects of the overall distributed DT architecture of edge-fog-cloud.

An earlier approach to the use of fog computing for real-time DT applications in [30] has shown the benefits of reducing response times when the DT is deployed in the fog node rather than the cloud. Another paper [31] introduces a fog computing extension to the MQTT IoT communication protocol for Industry 4.0 applications. By placing the MQTT broker at the fog layer, the approach enhanced data processing efficiency and reduced communication demands. In this work [31], the fog layer serves for prediction, acts as a gateway, and offloads complex processing tasks from the cloud to minimize latency and operational costs. The authors of [31] validated the architecture through energy consumption analysis and simulations, demonstrating its benefits compared to the traditional MQTT scheme in handling real-time data challenges posed by constrained IoT devices.

#### 2.2.2. IIoT Protocols and Middleware

Digital twins require a middleware protocol that serves as the connection between the physical entity and the digital entity. Many IoT applications leverage the use of such IoT protocols for communication with any other systems locally or remotely. ISO 23247 outlines this under part 4 (Networking View), which handles information exchange and protocols [8,32].

From the literature, we have identified the most used IIoT protocols with regards to digital twins [5] or fog computing architectures to be MQTT, OPC UA, AMQP, and CoAP, among others like DDS, MTConnect, MODBUS, etc. Among these, we reviewed these protocols in our experiments and adopted MQTT for its light weight and easy set up. 

#### 2.2.3. Microservices and DT Platforms

In terms of software stack for digital twin deployment, microservices are an important consideration with cloud computing platforms. For this work, an exploration of microservices and middleware deployment platforms suitable for our distributed DT framework was performed, focusing on open-source technologies. It was observed that not many researchers have explored this area. However, work by the authors of [33], documented microservices, middleware, and technologies suitable for DTs in smart manufacturing. 

The major consideration for the use of such microservice platforms is the ability to containerize applications or modules of the DT applications in a virtualized environment (Table 1). While in fog computing, the typical architecture involves the use of physical servers close to the data sources, it is possible to use the cloud or local hardware as a platform that distributes and splits the DT into modules that can be packaged in layers within a virtualized environment known as containers.

Docker (version 20.10.24) was considered among the options because of the nature of the project and experiments, making it easier for the deployment of the DT modules.

### 2.3. Digital Twin Architecture

Existing DT architectures mostly adopt a centralized deployment strategy, while some works show the importance of edge and fog computing in DT deployments. For instance, works like [19,30] showed the use of edge or fog as an improvement in response time, while [20] showed the use of edge computing for enhancement of data modeling for a machine tool DT. In this work, we approach the distributed DT for the case study of predictive maintenance with a hybrid architecture utilizing both the edge-fog-cloud depending on the specific layer requirements. The closest to this approach was found in [18]. Another work on distributed DT was found in [36], where the authors align the activities of the production shop floor with a physical layer and the edge–cloud collaboration layer with local and global DT tasks handling real-time manufacturing data. The work of [36] also recommended microservices to support modular development.

The key advantage to distributing DT is classifying a smart manufacturing system into unit, system, or systems of systems [6,21], or, similarly, it is usually labeled as local, system, and global DTs [36]. This framework is designed to accommodate the need for data collection from lower levels (sub-components of the manufacturing systems) and pre-processing and processing of all data from a component or production line, and the overall manufacturing system/shop floor is modeled in the global DT. Different approaches to the deployment of DT are usually adopted depending on the requirements of the system. In the methodology described by [6] and extended in Figure 2, for the context of PdM, the relationship between the three layers of DT can be summarized as shown in Figure 2.

The design of a DT architecture can adopt typical SDLC approaches. The work by [37], for instance, utilized ISO/IEC 1588/42010 [37] to guide the development stages as well as establish the concept of components and their relationships. As a PHM tool, DTs need to solve some of the challenges of implementing PHM such as the lack of real-time assessment of remaining useful life (RUL) in an interoperable and decoupled approach [38]. To achieve lightweight and seamless integration, an edge digital twin was explored in [39].

As introduced in our earlier work [19], this framework of splitting DT based on components aligns directly with the edge-fog-cloud architecture. The work in [40] presents a framework with local and global nodes to enhance manufacturing decision making. Local nodes include the equipment’s digital twins and a predictive model based on machine learning, aiding in improved decisions. Various tools enable condition-based maintenance and fault detection. The local DT, enriched with machine learning, contributes to overall prognostics and health management. The global node aggregates data, interacts with manufacturing execution systems (MESs) for accuracy, and facilitates scheduling and optimization based on data from the global DT, MES, and performance indices. While some of these approaches show good results, work by [41] evaluated this systematically by showing most architectures neglecting modularity in terms of plug and play and other non-functional qualities outlined in ISO23247 [8].

In terms of using the distributed DT concept to improve prognostics in PdM applications, it is ideal to design a framework that considers the hardware and software requirements, standards, and prediction feedback loop [4,19]. The solution architecture introduced in [19] is leveraged in this work to achieve the results of improving prognostics of wind turbines.

The major contribution envisioned by adopting this hybrid architecture is that the cost of cloud infrastructure can be significantly reduced in the long term if some functionalities of the DT solution are handled on-prem. Operational SCADA systems can be equipped with additional compute nodes (i.e., fog nodes) that can immediately handle the feedback functionality, which also involves running models to identify short-term dependencies and simulate the effect of immediate changes to the operations of the turbine components. The architecture is summarized below.

Edge/IoT Device Layer: the lower layer deals with unit-level DT, acquiring data from individual components such as the gearbox and pre-processing them through data cleaning and transmission to the upper layer.Fog Layer: the middle layer handles the system-level monitoring and feedback mechanism on prediction from the ML algorithms in real time. This is the layer where the middleware and microservices of the DT are also utilized.Cloud Layer: this layer deals with monitoring of global-level systems of systems, for example, the whole wind farm in our case study, training and retraining using historical data.

In Section 3, the theoretical framework of the layers is discussed in detail.

### 2.4. Research Gap

From the review of the literature, it was identified that a comprehensive digital twin for predictive maintenance in IIoT with a distributed architecture as in Figure 3, requires the key technologies in Figure 4 and needs key metrics to perform effectively. In terms of metrics, it mostly depends on the application used. However, our proposed solution aims to cover the following metrics, which all play a vital role in improving PdM digital twin implementation challenges [5], as well as extract a framework from the necessary key technologies for the predictive DT solution based on ISO 23247 [8].

Performance Metrics:Prediction feedback loop (DT to PT)AccuracyComputational latency

This framework also seeks to showcase the benefits of the proposed solution architecture in the scenario of handling multiple components, in our use case, multiple wind turbines in a wind farm. Each wind turbine has a collection of components that are key to its performance and whose health will be monitored in real time.

Another contribution of this framework is validating it with the scaling constraint (to multiple wind turbines), which has not been explored in the literature.

## 3. Theoretical Framework

The theoretical framework of the proposed predictive maintenance DT builds on the discussed concept of distributed DT for enhanced asset management using the edge-fog-cloud deployment strategy. This is validated with experiments and results, which are presented in Section 4.

### 3.1. Hypothesis

The proposed digital twin framework, which integrates cloud computing, fog, and edge computing, as well as relevant middleware and IIoT protocols, will demonstrate superior performance in terms of computational cost, response time, and accuracy compared to traditional centralized cloud-based DT architectures. 

This hypothesis assumes that the proposed framework will outperform traditional centralized cloud-based DT architectures, which have certain limitations in terms of computational cost and response time due to their reliance on a centralized infrastructure. The hypothesis also assumes that the proposed framework will provide accurate predictions for the maintenance needs of industrial components by leveraging IIoT protocols and edge computing capabilities with the constraint of handling multiple assets in real time.

### 3.2. Architectural Framework

#### 3.2.1. Layer 1: Edge Devices Layer

The first layer is the physical asset layer, which includes the industrial asset, sensors, and data acquisition devices. The sensors collect data from the physical asset and send them to the second layer.

Data Acquisition Sub-Layer: this layer is responsible for collecting and pre-processing real-time data from sensors and IoT devices.Sensors and Actuators Sub-Layer: along with sensors that collect data, the actuators that will receive control commands from the upper layers are also in this layer.

#### 3.2.2. Layer 2: Fog Computing Layer

The second layer is the data processing layer, which includes the fog computing nodes. This layer processes the data collected from the sensors and generates insights into the condition of the physical asset over the short term. This layer is responsible for storing and processing smaller volumes of data at the edge of the network, closer to the physical asset. The fog computing nodes are responsible for processing data in real time and providing fast responses to the physical asset.

Data Storage Layer: This layer is responsible for storing the pre-processed data in a distributed data store such as a Hadoop Distributed File System (HDFS), Cassandra, MongoDB, or influx DB, as they all support distributed processing. However, influx DB was selected because it supports real time seamlessly.Data Processing Layer: This layer is responsible for processing the pre-processed data to generate insights that can be used to train the predictive maintenance model. This layer can be implemented using technologies such as Apache Spark, Flink, or Hadoop MapReduce.Feedback Loop Layer: This layer is responsible for capturing feedback from the DT to the physical twin and using it to improve the predictive maintenance model. This layer can be implemented using technologies such as Apache NiFi or StreamSets.

#### 3.2.3. Layer 3: Cloud Computing Layer

The third layer is the cloud computing layer, which includes the digital twin and the predictive maintenance algorithms for all the components and the entire asset estate leveraging scalable storage and compute. The global digital twin, serving as a virtual replica of all the physical assets, is used to monitor operational statuses and simulate behavior to predict performance. The predictive maintenance algorithms use the data collected from the physical asset across the entire IIoT domain to train, retrain, and fine-tune the digital twin to predict when maintenance is required and to optimize the maintenance schedule.

Data Storage–Cloud Layer: This serves as the data archiving system that stores all the data from the IIoT systems throughout its lifecycle.Machine Learning Layer: This layer is responsible for training the predictive maintenance model using the insights generated by the data processing layer. This layer can be implemented using technologies such as Scikit-learn, TensorFlow, and PyTorch. Depending on the ML model implemented, these ML packages were used for the predictive maintenance algorithms.Model Deployment Layer: This layer is responsible for deploying the trained model in a distributed environment to make real-time predictions. This layer can be implemented using technologies such as Kubernetes, Docker Swarm, or Apache Mesos. However, to support our architectural framework and experimental platform, we used Docker at the fog and cloud layer to deploy the ML models.

### 3.3. Framework Standardization

The ISO 23247 [8] standard framework is a four-part framework that outlines a framework for digital twins in manufacturing. This work explored the relevance of using ISO 23247 [8] towards standardizing the framework. In a sequence of steps, all components and layers of the architectural framework were implemented using ISO 23247 as a guide. Similar approaches from [6,20,32] were adopted and extended. ISO 23247-2 [8] highlights the reference architecture in four sections, which decompose and link to all other parts of the ISO 23247 framework. 

Observable Manufacturing Element (OME) domain: Context for the physical twin (each wind turbine component), which is the basis of the DT. This interacts with DT interfaces for data collection and device control–feedback mechanism.Data Collection and Device Control Domain: this connects the physical twin (OME) to its unit DT through sensor data collection, synchronization, and actuating feedback to regulate operational conditions with decisions from the DT.Core Domain: this domain handles all DT services from analytics and simulations to feedback and user interaction.User Domain: this is the application layer through which users access the DT and see results through visualizations and other functionalities.

Figure 5 describes a mapping of the ISO 23247 reference architecture [8,32] to the presented case study of a wind turbine’s predictive maintenance distributed DT and how it links to the layers of the framework. This is an excerpt that guides the proposed framework in this study from the more detailed functional view of the ISO 23247 reference model for manufacturing.

The procedure adopted in the overall utilization of the standards outlined in the ISO 23247 can be described as a bottom-up approach as follows:Selection of standards for sensor interfaces, data collection and processing.Selection of interfaces for device control and handling of feedback from DT to PT.Selection of communication protocols and middleware.Selection of technology stack for representation of DTs such as JSON, DTDL. and other software implementation frameworks.Selection of deployment platforms based on specific data and processing requirements.Selection of functional services platform for visualization interaction with users via ERP, CAD, CAM, or others.

## 4. Methodology

This section breaks down the methodology used to achieve the framework. In summary, Figure 5 and Figure 6 give an overview of the methodology used to implement the DT solution. From Figure 6, the “SS_Cloud” highlighted in blue denotes operations performed in the cloud layer by the system of systems DT module, the “S_Fog” highlighted in green denotes the operations performed in the fog layer by the aystems DT modules, and finally the “U_Edge” highlighted in yellow denotes operations performed at the edge/IoT devices layer by the unit DT module.

### 4.1. System Architecture

The proposed architecture of using a distributed paradigm with edge/fog nodes is implemented using raspberry pi devices. The raspberry pi is connected via Wi-Fi to a “Cloud” PC. 

I.Components: The components were selected based on a review by NREL showing the most failure-prone components of a wind turbine. This was used to select two of the components for this work. These are the generator as system DT and its sub-components as unit DTs and gearbox as system DT and its sub-components as unit DTs.II.Software Architecture: The digital twin was developed using the Digital Twin Definition Language (DTDL) based on the Asset Administration Shell (AAS) to replicate the relationship between entities, and the operations of the DT such as the predictive maintenance models, alerts and fault classification algorithms for feedback operations. Figure 7 describes the modeling approach [42].

### 4.2. Metrics

These are key metrics evaluated to showcase the performance and benefits of the system. A point of emphasis is the fact that the proposed framework focuses on providing a solution that can enhance the deployment of digital twins in IIoT, which can improve the efficiency of asset management. The key metrics being recorded in the experiments are:I.Accuracy of the model to support PHM with respect to the real-time scenarios of turbine operation.II.Prediction feedback loop (DT to PT): how the computational platform, whether edge or cloud, supports the overall aim of the framework: data collection, pre-processing, and prediction feedback.III.Computational latency: time it takes for model run and feedback.

For the above metrics, the implemented predictive maintenance algorithms were monitored to evaluate their performance in the DT environment. Insights are aimed at answering the question “How best Digital Twins can be developed to achieve higher efficiency”. These questions are mainly:Does the distributed DT framework improve the effectiveness of a predictive maintenance solution?Does applying the standardization for DT show relevant improvement to the PHM solution?

### 4.3. Software

The relevant predictive maintenance models identified in the literature have been implemented and used as a benchmark. More algorithms are being developed. The DT Software developed uses the DTDL with communication interfaces that serve as middleware. These were implemented in the solution leveraging TCP/IP, Telegraf, and Mosquitto MQTT broker. The software script for the feedback mechanism is implemented in python (version 3.11.2). The simulation of WT components in software is based on the acquisition of sensor data, publish/subscribe leveraging MQTT, and data queries from the real-time database influx DB. This follows the model described in Figure 8.

#### 4.3.1. Machine Learning Algorithms

As identified in the literature, some of the machine learning algorithms used for the data-driven predictive maintenance approach, and which have shown good results [15], were implemented. In all algorithms used, the input and output variables were adopted from works in the literature [15], described in Table 2.

The algorithms are described below.

Multiple Linear Regression (MLR): This algorithm uses scikit learn to model the relationship between the inputs and the output by fitting a linear equation.Long Short-Term Memory (LSTM): This is a version of a Recurrent Neural Network (RNN) that makes its predictions by using the order sequence of data to learn the termly dependencies of data. This is why it is suitable for IoT time series predictions.XGBoost: This algorithm uses decision trees for gradient boosting and works by combining weaker learners to create a stronger learner. It is considered one of the best algorithms for time series predictions and hence why it is suitable for IoT data.

The application of the model pre-processing and processing through the utilization of the Digital Twin data is outlined in Algorithm 1.
**Algorithm 1:** Model Pre-processing and Training

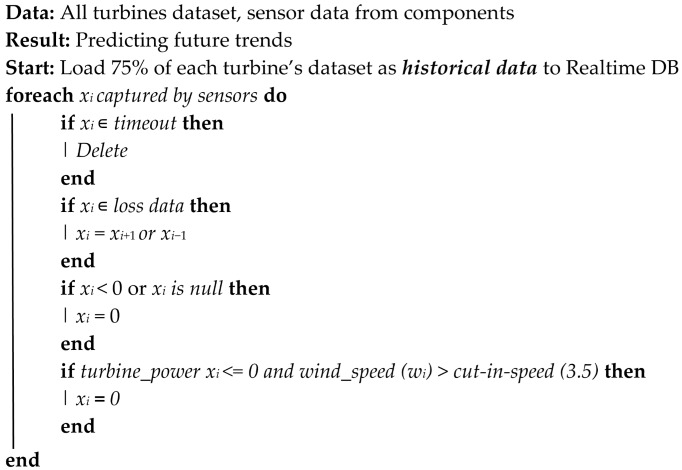


Identify variables of interest for each component and assign them as input and output variables.Split historical data into 80% train set and 20% test set.Forecast the future trend using MLR, XGBoost, LSTM, SVM, Random Forest (then select best model).

#### 4.3.2. Fault Thresholds

Furthermore, the post-processing is achieved by using the statistical process control (SPC) called Stewhart Process Control [15] given by the formula in Algorithm 2. The deviations are the differences between the actual and the predicted values, from which the moving range formula is applied followed by the calculation of upper and lower control limits to find deviations that indicate an anomaly in the normal wind turbine operation. This is described in Algorithm 2.
**Algorithm 2:** Model Post-processing

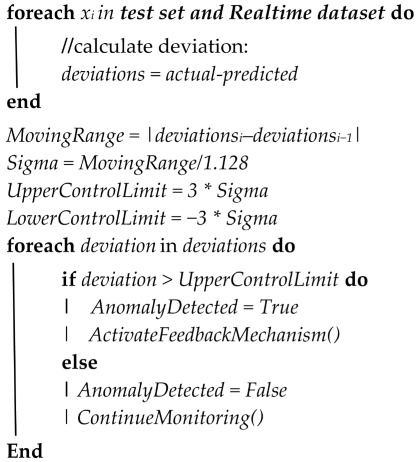


#### 4.3.3. Feature Extraction

To further understand the effect of each of the variables on the performance or health of each component, feature importance ranking was performed as a form of sensitivity analysis used to rank the variables based on the influence on the machine learning models. This helped us in identifying the key sub-components to focus on for the feedback mechanism. This was a form of validation from the literature that supported this methodology, as outlined in Table 2 above.

#### 4.3.4. Prediction Feedback 

This serves as the script that coordinates the feedback on prediction of failure. The relevant sub-components are regulated through simulations to handle the effect of their current behavior on the health of the component being monitored by the digital twin. These components are identified by the sensitivity analysis as described in Section 4.3.1 above. This can be described by Algorithm 3.
**Algorithm 3:** Feedback Mechanism

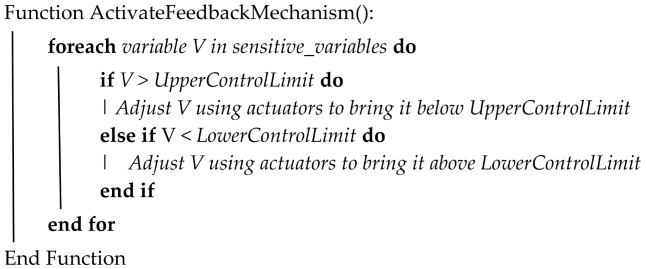


### 4.4. Data Platform

The datasets serve as sources of data from the components. Influx DB is used as the real-time database that streams data. Each turbine out of the four turbines has a bucket in influx DB. In one instance, these buckets hold 75% of the data (that is 18 months’ worth of 10 min readings), and the remaining 25% are used for real-time streaming and prediction by the DT. This was reconfigured in different instances of the experiments to model and understand the behavior. For easier analysis due to time constraint, each 10 min reading is streamed every second.

### 4.5. Connectivity Middleware

The experiment set up was implemented with Wi-Fi as the network connection and connectivity, MQTT, and Telegraf as middleware, and this is between the raspberry pi at the edge and the fog/cloud pc.

This middleware is used to support stream processing that supports the simulation aspect of the digital twin in real time. An example of this is showcased in Section 6.4.2 of this paper, where a subject matter expert may wish to reduce operational status of a certain asset based on predicted fault.

### 4.6. Functional Requirement

This is the final phase of the experiment set up that provides an interface to the digital twin to highlight its performance using a simple application such as a web platform. The functional services which relate to the user entity section of the ISO 23247 is beyond the scope of this work. The digital twin proposed in this work also serves as the core entity of the ISO 23247 reference architecture, and this handles the management and monitoring of the predictive maintenance DT. A description of how the machine learning model, as well as the simulations, were handled by the DT can be summarized by Algorithm 4.
**Algorithm 4:** Model Handling in DT

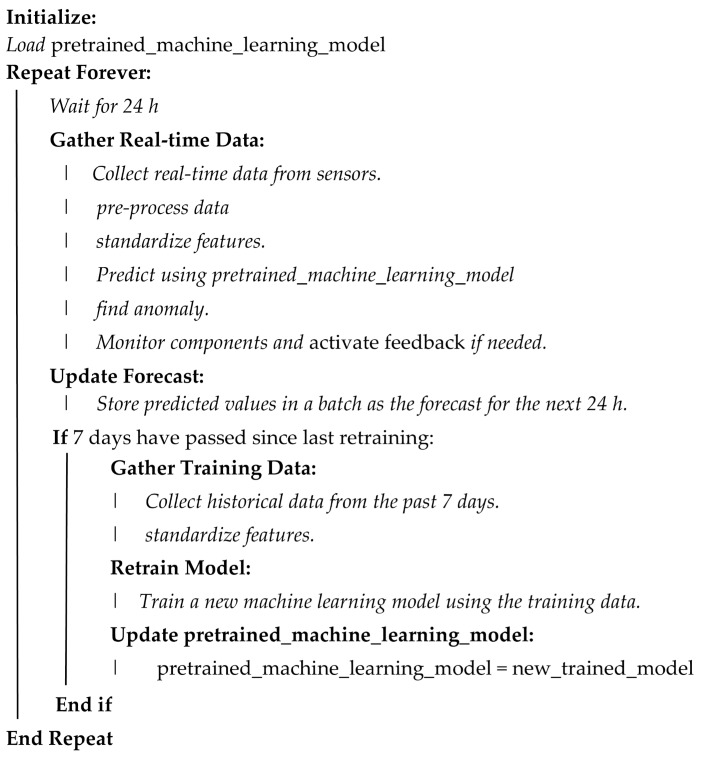


## 5. Experiment Set Up

The experiment for the solution was achieved using some hardware and software along with a dataset from EDP [43]. The overall setup is described in Figure 9, which is an implementation of the solution architecture introduced in Figure 3 [19].

I.Edge IoT Devices: this layer is equipped with raspberry 3 devices (64-bit @ 1.4 GHz 1 GB SDRAM) that simulate “sensors”, which publish real-time sensor data using MQTT brokers, from each wind turbine component, and receive feedback from the upper layers DT.II.Fog Nodes Layer: This layer is equipped with a fog node serve/pc (Lenovo IdeaCentre Mini PC 1.5 GHz 4 GB RAM 128 SSD). This aggregates the components for each turbine using containerized microservices, Docker pushes the real-time sensor readings to influx DB buckets in the fog (batch data streaming point will be good here) for short-term storage and the cloud for longer-term storage. This layer also hosts the system DT (DT for each turbine). The script, described in Algorithm 3, that regulates the components’ behavior once faults are predicted is also hosted in this layer. Both batch and real-time data collected from the sensors are processed in this layer. All the activities highlighted later in the feedback section relate to this layer.III.Cloud Layer: This layer is equipped with a higher computational capacity, using a PC (Intel Core i5 CPU @ 3.30 GHz 8.00 GB RAM 64-bit OS). This layer hosts the global DT, training, and testing set of all the ML models for each component and periodically retrains the models as more data and faults are identified over time using the longer-term historical data in the influx DB.

A closer look at the overall workflow from the experiment is described in Figure 10.

## 6. Results and Discussion

The result of this work follows the earlier work performed in the literature as benchmarks [14,15,17]. This is to showcase the relevance of the distributed digital twin framework, as well as apply standards towards improving asset management in manufacturing systems—specifically predictive maintenance of wind turbines in a wind farm. This section starts with highlighting the failure logs outlined in the dataset [43] used for the experiments in Table 3. This is followed by an analysis on the performance of the implemented models in two scenarios—(1) adopting a cloud-centralized strategy and (2) applying the distributed DT framework as a strategy that enhances real-time feedback leveraging the distributed DT. This will be followed by the impact of the distributed framework in enhancing two-way feedback leveraging the fog layer of the framework.

### 6.1. Model Pre-Processing

For all the Machine Learning Models outlined in Section 4.3.1, the first stage after loading the historical data from the specified influxdb bucket of each wind turbine hosted in the cloud-PC, is to pre-process the data by cleaning and removing outliers based on Algorithm 1. This is followed by the identification of the specific inputs and output needed for each component as outlined in Table 2.

Once the data is cleaned, and made reasonable for a WT operation scenario, it is then split into training and test set. The data was split into 70% train set and 30% test set for each year 2016 and 2017, separately.

### 6.2. Model Processing

The pre-processing stage was followed by the processing of each of the models for each of the components. In the processing stage, parameters for each of the models are configured before training is done. For instance, the architecture of the LSTM model for the gearbox is different from that of the Generator. This is one of the functions of the DT in accommodating each component and model based on its specific requirements. While all the implementations share similarity, this feature is important in making the DT handle each component with more detail, with a dedicated mode/DT for enhanced accuracy. Table 4 gives a summary of the performance, in terms of accuracy of all the components categorized by Turbine, Year of Operation, Component and Algorithm used.

#### 6.2.1. Gearbox

From the result metrics, when taking a closer look at the initial accuracy metrics for the gearbox component overall training and testing, each year performed individually, it was observed that the models for both Turbines T01 and T06, taken as an example, behaved similarly where there was an increase in RMSE and MAE from the 2016 to the 2017 data, and, in both cases, MLR and XGBoost outperformed LSTM. However, an interesting point to note is that for LSTM, there was a decrease in RMSE and LSTM between 2016 and 2017, with that of 2017 having better performance significantly. This showcases the relevance of LSTM as a deep learning approach that supports termly dependencies, suggesting that, for this case study, seasonality, in terms of the four quarters/seasons of the year, needs to be considered. This is due to atmospheric and weather conditions that affect temperature, wind, and other environmental factors. 

This finding highlights the point that the training and testing of the model need not be a standard ratio in all cases; as such, the distributed DT having short-term and long-term storage in the fog and cloud layers, respectively, allows for handling this requirement.

#### 6.2.2. Generator

For the generator bearing component, the results show a different pattern. Firstly, the RMSE and MAE, which indicate how far off or close the predictions are to the actual values of the generator bearing temperature, are higher because the generator bearing operates at a higher temperature than the gearbox bearing. This was also easily identified by the DT. In terms of the accuracy metrics, the results show that LSTM and XGBoost outperformed MLR in T01, while XGBoost and MLR outperformed LSTM in T06.

#### 6.2.3. Further Analysis

From analyzing results from the behavior of all the turbines, T01, T06, T07, and T11, across both years and all the models, it was found that the gearbox has more susceptibility to the seasonality factor than to the generator. This is also due to the fact that for a gear-type wind turbine, which is the case in this study, the gearbox is closer to the inputs, which transfer the mechanical energy to the generator as output [44]. For the models, while RMSE and MSE were used for accuracy, the lower the values, the better the accuracy. However, the R-squared score, which is between 0 and 1, remains within the same range for all components, and having higher values (close to 1) indicates the models performed well.

Additionally, while one algorithm performs well for a certain component, another algorithm may perform better for another component, as can be seen in the plots in Figure 11a–d. This fact supports this work’s proposal on why the distributed DT framework is relevant as a tool for suitable experimentation and PHM fine tuning. Further discussion on this is in Section 7 of this paper.

### 6.3. Model Post-Processing

The next stage in our methodology when the models are trained and tested is to select the best-performing algorithm among the tested algorithms. As mentioned in the previous section, while all algorithms performed well, the best algorithm to work with is subject to the component and often the training period due to the seasonality observed in the wind turbines behavior. However, considering all the above, for the purpose of these experiments, it was observed that among all the algorithms, XGBoost performed best in most cases. As such for the stage of post-processing failure prediction and feedback, XGBoost was adopted, and the next stage of the methodology was achieved. Figure 12 and Figure 13 show the actual vs. predicted of Turbine T06—Gearbox and T07—Generator towards the time of failure of the components.

#### 6.3.1. Failure Prediction—Gearbox

After prediction with the acceptable algorithm and accuracy, the next stage was to apply the statistical process control formula as outlined in Algorithm 2. The failure data from Table 3 were used to look out for failures in the wind turbine operation around the specified date when the component fails. An alert was configured in the DT to monitor the threshold and pick up a deviation, which will be the point of failure prediction. Figure 14 shows how a failure in Turbine T06’s gearbox, which happened at time 2017-10-17 08:38, was picked up a week before, on 2017-10-11 00:30.

#### 6.3.2. Failure Prediction—Generator

Similarly for the generator, once prediction of the operations of the generator is performed, the deviations are detected and SPC applied to predict failure. However, the generator uses a higher lower and upper control limit due to its behavior. As shown in Figure 15, failure was predicted by the DT on 2017-08-10 00:00 with multiple alerts, which were almost two weeks before the actual failure occurred, as highlighted in Table 3.

### 6.4. Prediction Feedback Loop

This section highlights the implementation of the prediction feedback loop from the digital twin to the physical twin (PT). The major aim of this metric in our framework is to highlight the relevance of the distributed DT in accommodating the requirements for a modular DT that handles the feedback from the moment a failure is predicted by the overall global model in the cloud layer. The DT node specific to the component will then activate a model that evaluates how a behavior change in the operation of the component can improve the remaining performance (time) before failure. This is envisioned to optimize the utilization of the asset. After reviewing the 1-month behavior before failure (Figure 16, Figure 17 and Figure 18) and the feature influence ranking in Figure 19, Figure 20 below shows how this distributed DT handled the failure of T07 and T06 as indicated from their failure in Section 6.3.1 and Section 6.3.2 above.

#### 6.4.1. Pre-Failure Assessment

To achieve the prediction feedback loop, the DT module handling each component must triage the deviations from the moment of failure prediction and perform some predictive modeling on them. In our case study, from the EDP [43] dataset, taking turbine T06 2017_Gearbox as an example, the prediction shown in Figure 14 was on the 12th of October 2017 08:38. However, actual deviations that indicated a potential issue with the gearbox started occurring two weeks before the failure, on 5th of October 2017 at 12:07. Figure 16, Figure 17 and Figure 18 show the behavior of the gearbox temperature in a weekly snapshot up to 1 month before failure. It can be observed that within this 1 month before failure, the first two weeks (20 September to 4 October 2017) appear to be normal, with their SPC within the acceptable range, while the second two weeks (4 October to 18 October 2017) show the deviations that predicted failure.

**Figure 16 sensors-24-02663-f016:**
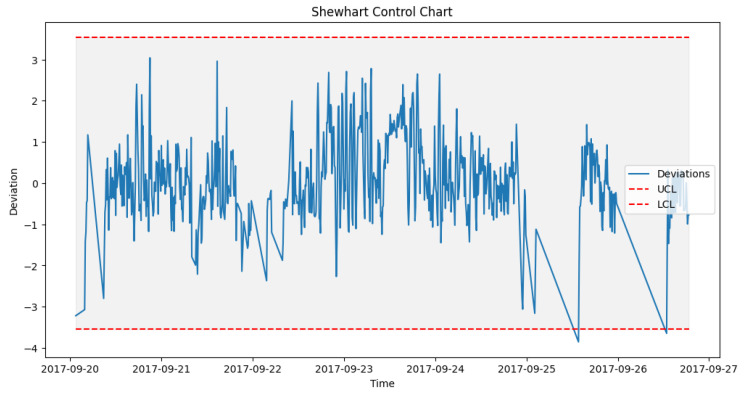
T06_Gear control chart 20 September to 27 September 2017.

**Figure 17 sensors-24-02663-f017:**
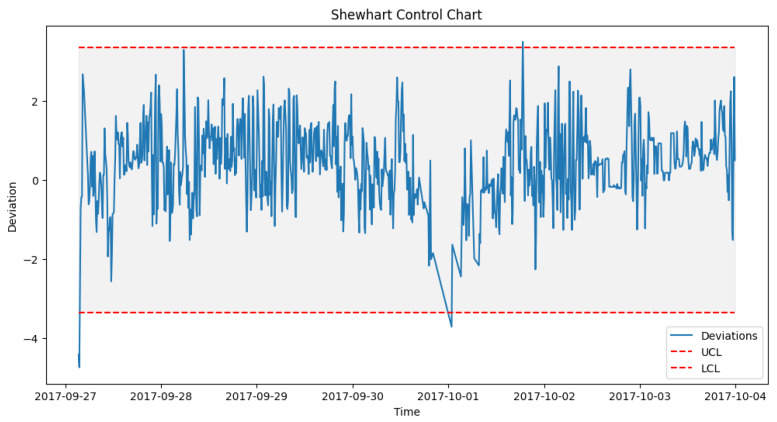
T06_Gear control chart 27 September to 4 October 2017.

**Figure 18 sensors-24-02663-f018:**
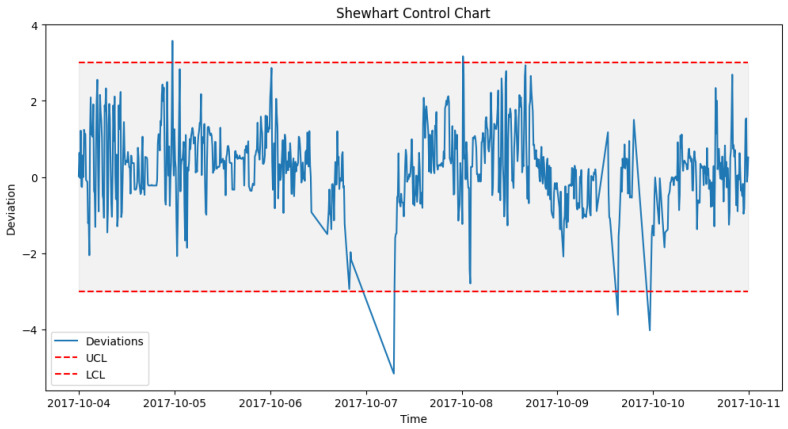
T06_Gear control chart 4 October to 11 October 2017.

As seen in Figure 14 from Section 6.3.1, which shows the behavior for 11 to 18 October, in which our actual failure was recorded on 17 October at 8:38.

#### 6.4.2. Feedback

Having established that our model detected deviations from normal operation since 5 October 2017 at 12:07, we now show how the feedback mechanism, as described in Algorithm 3, simulated the DT to PT two-way feedback by modeling the regulation of some components.

To achieve the feedback simulation, it was necessary for the DT module to identify the top most important features from the latest model run. Figure 19 shows how the gearbox oil temperature, rotor RPM, and nacelle temperature had the most influence on the gearbox temperature model prediction. This was achieved by simply identifying the variables with the highest absolute coefficients on the regression model’s output, given by
y=b0+b1x1+b2x2+⋯bnxn
where y is the predicted output (gearbox temperature), b0 is the intercept, and b1, b2 … bn are the coefficients associated with each input feature (x1, x2 … xn), i.e., gearbox oil temperature, rotor RPM, etc.

**Figure 19 sensors-24-02663-f019:**
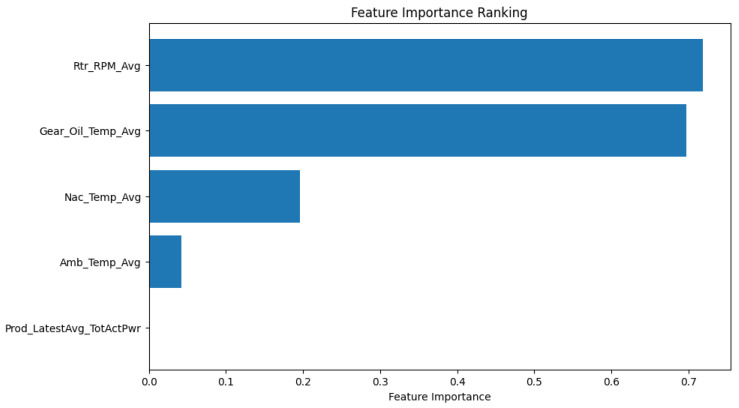
Feature importance ranking for T06 gearbox.

Based on this, we selected the top three features that are most sensitive to the gearbox behavior and implemented feedback simulations based on this.

Figure 20 shows the behavior of the gearbox after the simulation where the feedback loop handled the failure by identifying the components most sensitive to the original model that predicted the failure and reducing them by 30%.

**Figure 20 sensors-24-02663-f020:**
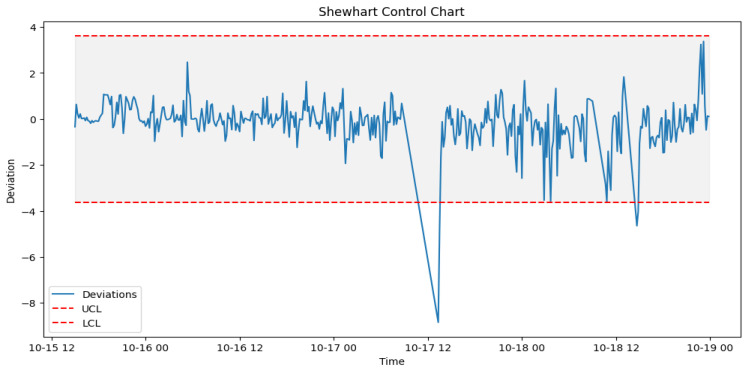
SPC operations using DT feedback mechanism—30% reduction example.

#### 6.4.3. Accuracy

From the metrics of the performance of the model during the feedback runs of the model, we compare the actual T06_2017_Gear performance metrics and those of the feedback run performance in terms of RMSE, MAE, and R-squared (Table 5 and Figure 21).

This comparison in Figure 21 shows that despite the splitting of the DT, where each module handles a component in the fog in the constrained real-time requirement of handling, the failure is acceptable and offers an improvement.

From our results, we can infer that distributed architecture can strategically utilize on-premise or distributed compute nodes for real-time fault prediction and handling, optimizing latency and cost-effectiveness near data sources. Cloud infrastructure complements by handling intensive computational tasks for long-term analytics, like training deep learning models on historical data. This balanced approach ensures effective predictive maintenance, leveraging both short-term responsiveness and long-term analytical capabilities. Although beyond the scope of this paper, containerization using Docker can enhance computational efficiency within distributed systems, reinforcing their suitability for predictive maintenance applications.

## 7. Discussion

This work presented a framework that supports the implementation of a distributed digital twin capable of improving PHM in an IIoT-enabled manufacturing setting such as wind turbines. From the architectural framework, methodology, and results, it is evident that this presented DT framework can also help both in real-life PHM solutions as well as in experimentation. In the framework, different components’ operational readings were collected in real time using sensors simulation, then, based on historical and real-time data from the DT, different machine learning techniques were applied to predict potential issues and failure of a component. In terms of the versatility of the architecture, it supported, for example, which algorithm was more suitable for which component, as well as which model was best in handling seasonality.

For instance, when we applied LSTM for the gearbox and generator of the turbines, it was noted that both components behaved differently in terms of the accuracy of the predictions when parameters of the LSTM architecture are fine-tuned differently.

The potential benefit of adopting this framework is that the DT can seamlessly allow researchers, engineers, and operations teams to simulate and apply the best model and parameter fitting, whether for model training, testing, or even in real time as the DT framework supports the prediction feedback loop. The concept of Intelligent DT in this framework suggests that the DT could automatically evaluate these reconfigurations and feedback based on the behavior of the component being managed, and in real time. 

For architectural considerations, while the Global DT handles the model training and retraining in the cloud layer, the feedback handling model/PHM solution configurations are handled in the fog and edge layers. This shows the relevance of the distributed DT framework presented in this study.

Finally, with failure predictions achieved at least two weeks in advance, the DT predicted the behavior of the gearbox temperature based on some selected variables, and the feedback mechanism attempts to handle this by regulating the deviations and controlling the identified components to stay within threshold, potentially extending the usage before failure of the asset. This shows that the proposed architecture supports our objective of handling PHM issues in real time. As with PHM, the application subject matter expertise (SME) to evaluate, monitor, and improve the models will be an iterative process. As such, this feedback loop in the fog layer can automatically simulate several scenarios in real time (e.g., running many variations of the sliced model) and instantly select the best control strategy to improve performance, for example, this could be to reduce RPM by 20% instead of by 30%, or which component to regulate from the sensitivity analysis outcome.

### Limitations

The research design was aimed at establishing the benefits of the proposed architecture. However, the results in this paper highlighted two out of the three metrics proposed, specifically the computational latency aspect of the experiment has not been discussed in detail. Additionally, the scalability has not been factored in, as data of only four wind turbines were available.

## 8. Conclusions and Future Work

This study presented a distributed digital framework towards improving asset management of wind turbines as a case study for applying a more standardized framework within IIoT where predictive maintenance and DT can be key technologies for achieving improved business outcomes. The framework showed the utilization of the distributed DT architecture to apply a prediction feedback loop from the DT to the PT, improving impact by predicting failures as early as possible and remediating them with acceptable accuracy (model performance), and the provision of a computational platform that offers better utilization of computational performance in terms of latency to satisfy real-time requirements in the solution. With the utilization of on-prem compute nodes, the framework envisions lower cloud infrastructure costs. While the concepts have shown how the strategy achieves this, more research needs to be performed on exploring the machine learning techniques’ performance peculiarities with respect to predictive maintenance.

Further work in this study seeks to investigate the relevance and benefits of the framework in supporting transfer learning for integration of newer components within an existing IIoT infrastructure, among other important ML techniques that support asset management in IIoT.

## Figures and Tables

**Figure 1 sensors-24-02663-f001:**
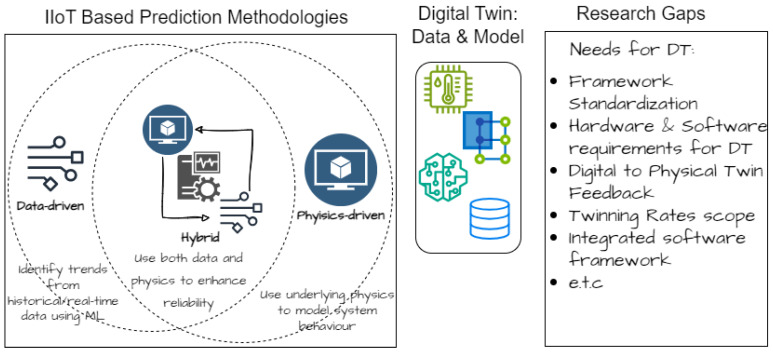
Predictive DT models and associated gaps in IIoT.

**Figure 2 sensors-24-02663-f002:**
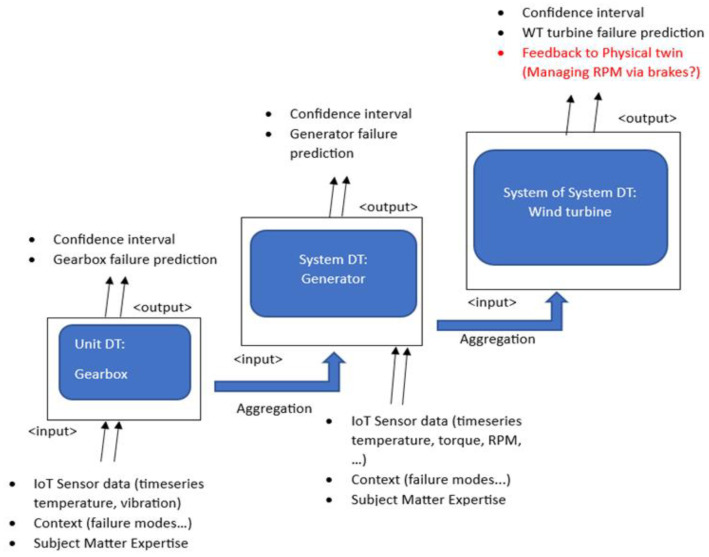
High-level hierarchy of a DT solution.

**Figure 3 sensors-24-02663-f003:**
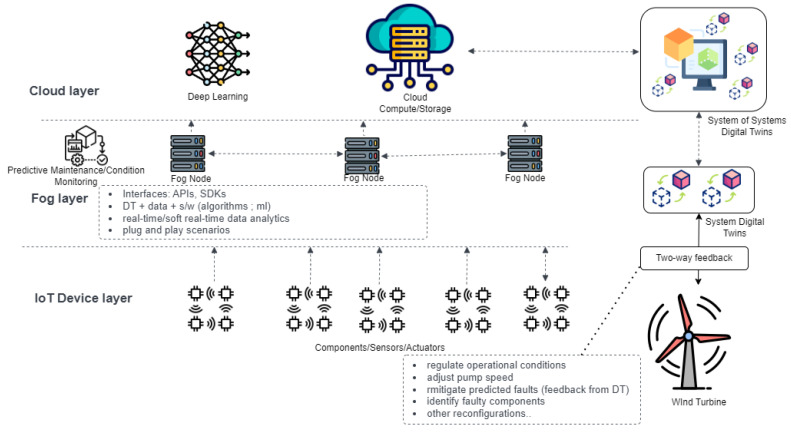
Distributed DT architecture.

**Figure 4 sensors-24-02663-f004:**
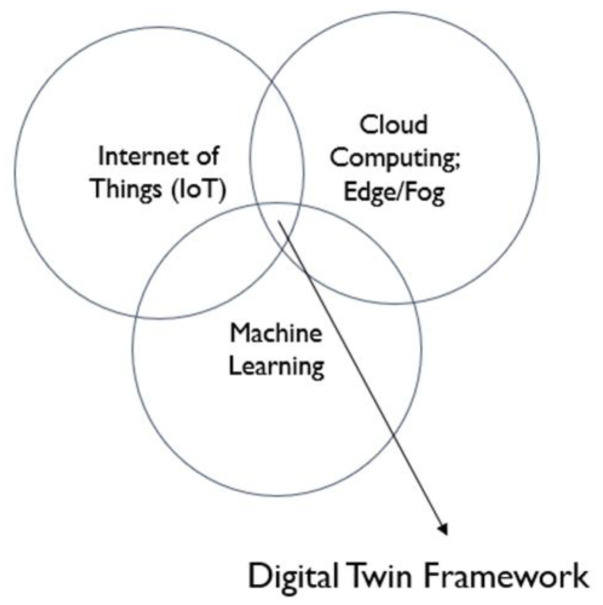
Key technologies for DT framework.

**Figure 5 sensors-24-02663-f005:**
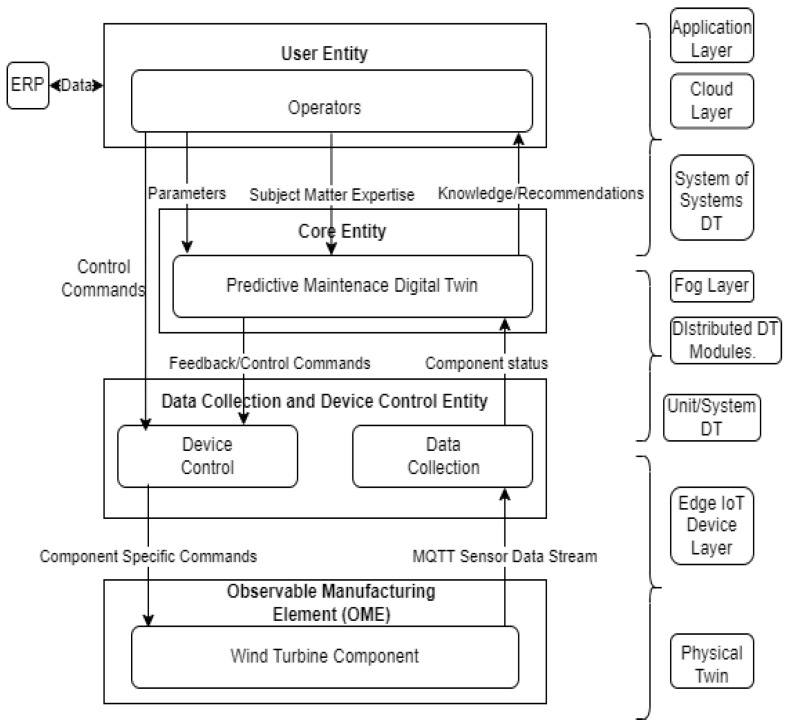
ISO 23247 reference architecture mapping.

**Figure 6 sensors-24-02663-f006:**
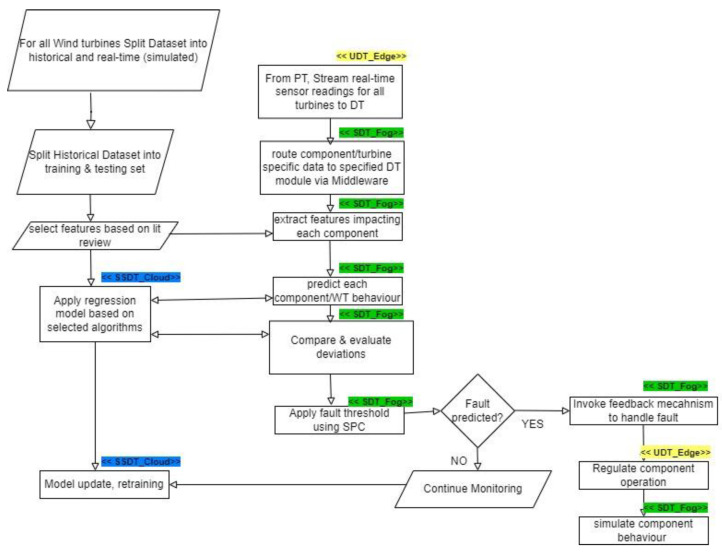
Block diagram of predictive DT solution.

**Figure 7 sensors-24-02663-f007:**
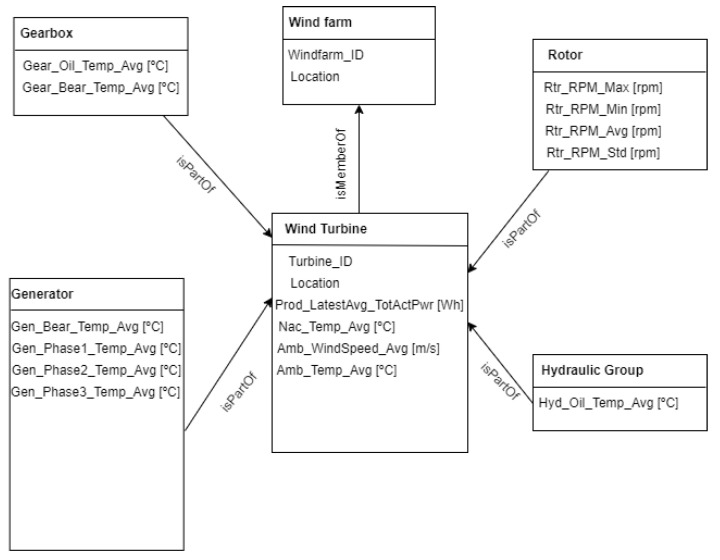
Data model for wind turbine DT based on EDP dataset.

**Figure 8 sensors-24-02663-f008:**
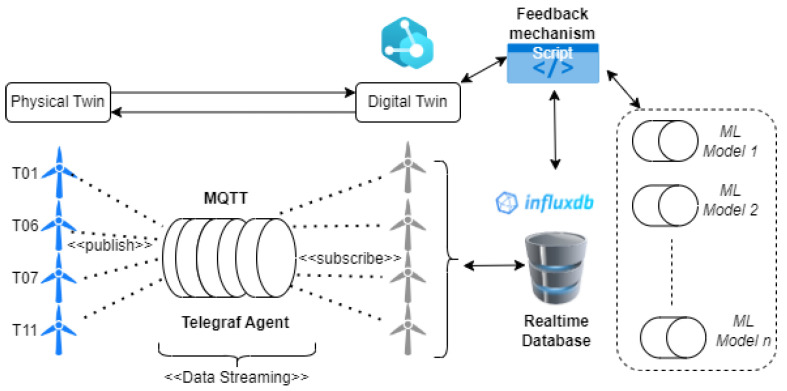
Overview of system architecture with middleware/streaming.

**Figure 9 sensors-24-02663-f009:**
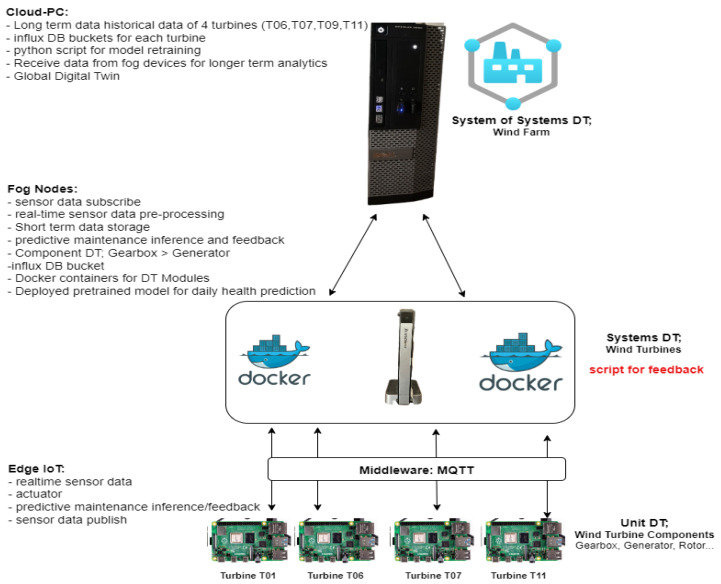
Experiment setup based on solution architecture.

**Figure 10 sensors-24-02663-f010:**
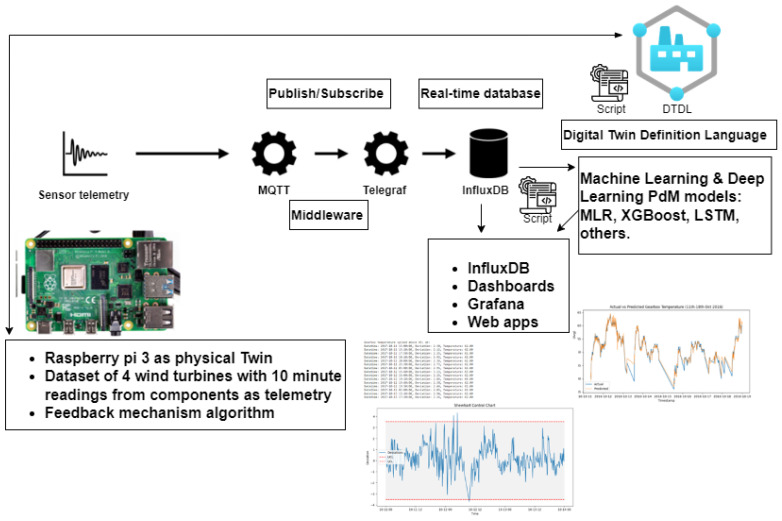
Functional application of experiment.

**Figure 11 sensors-24-02663-f011:**
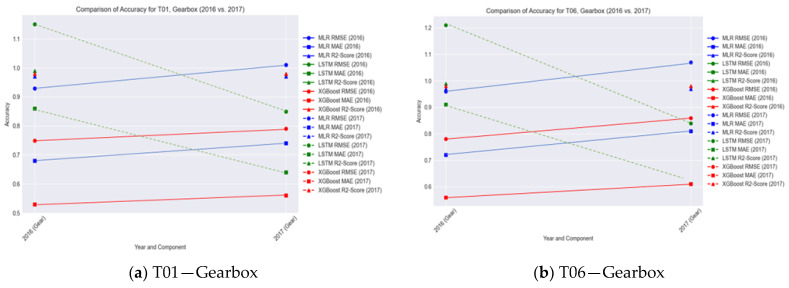
T01 and T06—2016 vs. 2017 accuracy comparison by component.

**Figure 12 sensors-24-02663-f012:**
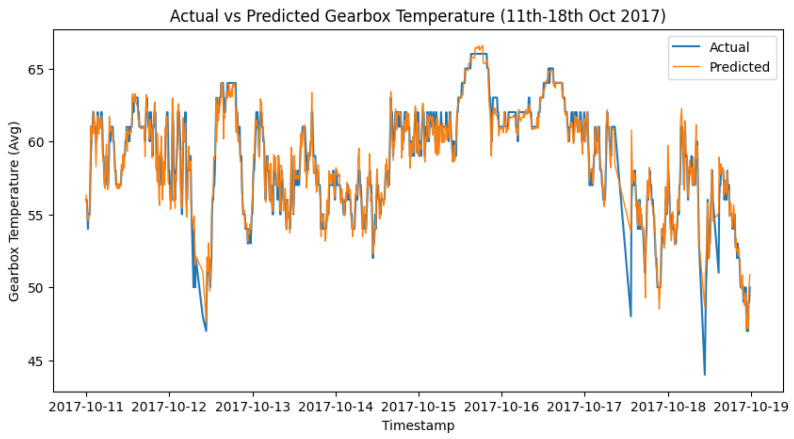
T06 gearbox 2017 actual vs. predicted.

**Figure 13 sensors-24-02663-f013:**
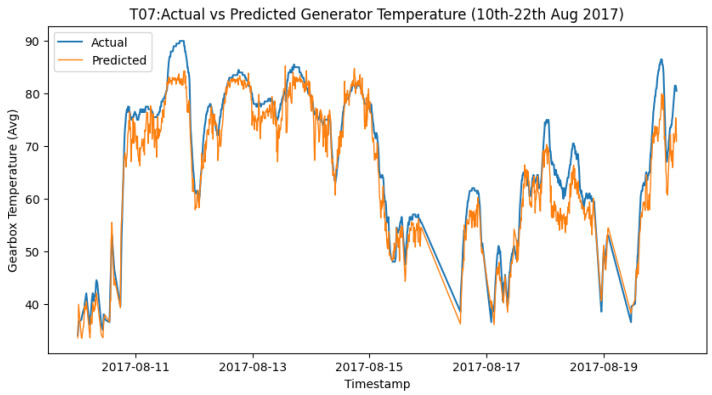
T07 generator 2017 actual vs. predicted.

**Figure 14 sensors-24-02663-f014:**
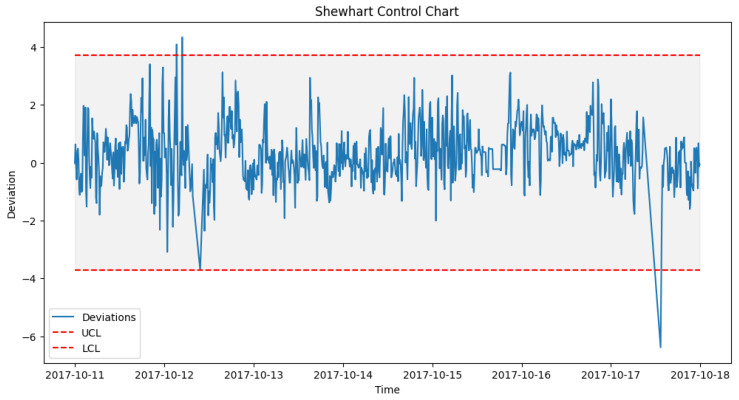
SPC for T06 gearbox failure prediction alerts.

**Figure 15 sensors-24-02663-f015:**
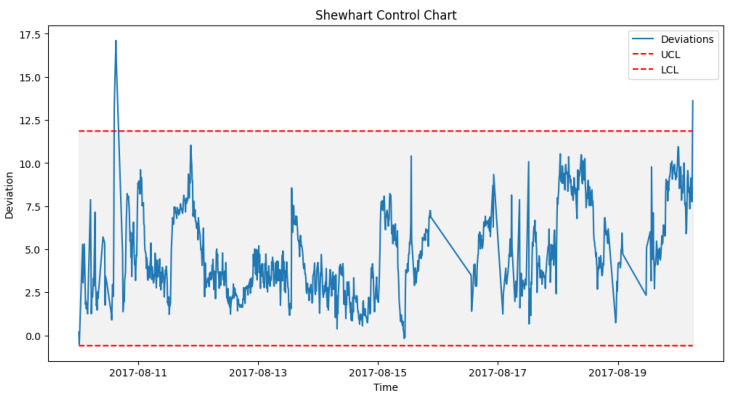
SPC for T07 generator failure prediction alerts.

**Figure 21 sensors-24-02663-f021:**
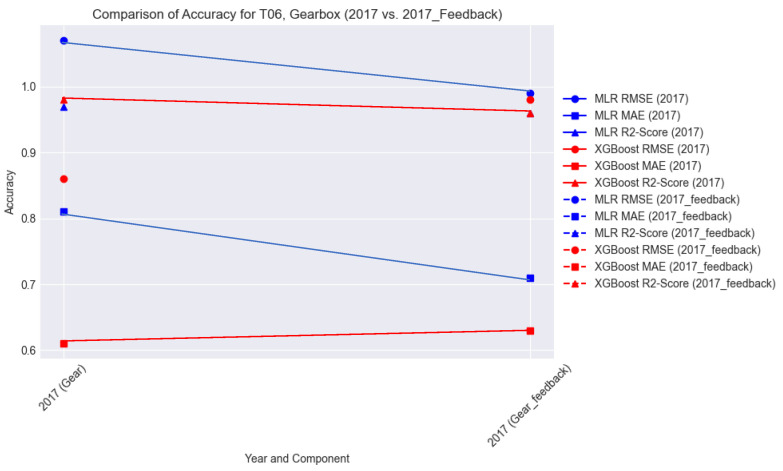
Accuracy of model before and during feedback run.

**Table 1 sensors-24-02663-t001:** Microservices for DT architecture.

Concept	Description	Tool
Containerization	Utilizing microservices architecture for loosely coupled, capabilities oriented, and packaged deployment software.	[33] Docker [34], Kubernetes
DT Middleware	Platforms supporting connectivity middleware, device, and data integration.	[33] Eclipse Kapua, Eclipse Kura, Eclipse Ditto [35]
Real-Time/Batch Stream Processing	Technologies supporting processing data with compute capabilities in batch or real time	[33] Apache Kafka, Apache Flink, Apache Spark [35], Apache Hadoop

**Table 2 sensors-24-02663-t002:** Selected components for predictive maintenance model.

Component	Inputs	Output	Ref
Gearbox	Nacelle Temperature	Gearbox Bearing Temperature	[15]
	Rotor Speed		
	Active Power		
	Ambient Temperature		
	Gearbox Oil Temperature		
Generator	Nacelle Temperature	Generator Bearing Temperature	[15]
	Active Power		
	Generator Speed		
	Generator Stator Temperature		

**Table 3 sensors-24-02663-t003:** Actual failure logs from EDP dataset.

Turbine	Component	Timestamp	Failure Type
T06	GEARBOX	2017-10-17T08:38	Gearbox bearings damaged
	HYDRAULIC_GROUP	2017-08-19T09:47	Oil leakage in hub
T07	GENERATOR	2017-08-21T14:47	Generator damaged
	GENERATOR_BEARING	2017-08-20T06:08	Generator bearings damaged
	HYDRAULIC_GROUP	2017-06-17T11:35	Oil leakage in hub
	HYDRAULIC_GROUP	2017-10-19T10:11	Oil leakage in hub
T11	HYDRAULIC_GROUP	2017-04-26T18:06	Error in the brake circuit
	HYDRAULIC_GROUP	2017-09-12T15	Error in the brake circuit

**Table 4 sensors-24-02663-t004:** Result metrics.

Turbine	Year	MLR			LSTM			XGBoost		
		RMSE	MAE	R^2^-Score	RMSE	MAE	R^2^-Score	RMSE	MAE	R^2^-Score
T01	2016 (Gear)	0.93	0.68	0.97	1.15	0.86	0.99	0.75	0.53	0.98
	2016 (Gen)	3.91	3.25	0.90	3.86	3.10	0.90	3.98	3.17	0.90
	2017 (Gear)	1.01	0.74	0.97	0.85	0.64	0.98	0.79	0.56	0.98
	2017 (Gen)	4.28	3.17	0.89	4.04	2.94	0.90	3.83	2.77	0.91
T06	2016 (Gear)	0.96	0.72	0.97	1.21	0.91	0.99	0.78	0.56	0.98
	2016 (Gen)	2.99	2.01	0.92	3.19	2.26	0.91	3.36	2.44	0.90
	2017 (Gear)	1.07	0.81	0.97	0.84	0.61	0.98	0.86	0.61	0.98
	2017 (Gen)	3.63	2.26	0.89	3.89	2.38	0.85	3.63	2.2	0.87
T07	2016 (Gear)	1.04	0.74	0.96	1.15	0.86	0.99	0.81	0.57	0.98
	2016 (Gen)	2.55	2.04	0.95	2.49	1.96	0.95	2.60	2.05	0.95
	2017 (Gear)	1.03	0.76	0.96	0.86	0.65	0.97	0.79	0.56	0.98
	2017 (Gen)	3.59	3.11	0.92	4.87	4.42	0.80	3.83	2.77	0.91
T11	2016 (Gear)	1.16	0.87	0.96	1.12	0.80	0.99	0.86	0.62	0.98
	2016 (Gen)	3.30	2.61	0.89	3.45	2.68	0.88	3.31	2.62	0.89
	2017 (Gear)	1.22	0.92	0.96	1.02	0.76	0.97	0.87	0.64	0.98
	2017 (Gen)	3.75	2.95	0.90	3.43	2.82	0.91	3.46	2.80	0.90

**Table 5 sensors-24-02663-t005:** Accuracy of model during feedback run.

Turbine	Year	MLR			XGBoost		
		RMSE	MAE	R^2^-Score	RMSE	MAE	R^2^-Score
T06	2017 (Gear)	0.99	0.71	0.96	0.98	0.63	0.96
T07	2017 (Gen)	3.29	3.02	0.87	3.64	2.43	0.89

## Data Availability

The data used in this research are from EDP as referenced in text.

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
