# Peer review of "Towards a Distributed Digital Twin Framework for Predictive Maintenance in Industrial Internet of Things (IIoT)"

_sensors, 2024, doi:10.3390/s24082663_

Round 1
Reviewer 1 Report
Comments and Suggestions for Authors
This paper proposed a distributed digital twin framework for predictive maintenance. Although the topic is meaningful, I cannot see great contributions of this manuscript for DT research community. Authors indicate a research gap in Sec 2.4, but I don't think it is a key limitation in DT-driven predictive maintenance. So the novelty of the paper is obscure and difficult to judge.
Reviewer 2 Report
Comments and Suggestions for Authors
This paper introduces a distributed digital twin framework based on cloud-fog-edge collaborative information processing, and conducts actual case verification using wind turbine internal SCADA data. It's interesting, but there are some issues that need to be addressed as follows.
1. What is the difference between the IIOT and the IOT? Is the digital twin construction method proposed in the paper based on the IOT or the IIOT? What is the relationship between the digital twin framework of this paper and the IOT or the IIOT?
2. What exactly does the digital twin in the cloud computing layer of the digital twin architecture refer to? How to build a virtual replacement for physical assets?
3. What exactly does the digital twin in the cloud computing layer of the digital twin architecture refer to? Predictive maintenance model for wind turbines? How does the architecture proposed in this paper build a virtual replacement for physical assets?
4. On what basis are the fault thresholds chosen in the gearbox and generator fault prediction section?
5. How is the 30% reduction in failures calculated in Figure 20?
6. What are the advantages of the distributed architecture proposed in this article compared to the centralized architecture? More discussion needs to be added in the results section.
7. Can the signal transmission delay of the digital twin framework based on distributed architecture meet the real-time requirements for wind turbine predictive maintenance? Offshore wind power is an important part of wind power technology. How applicable is the method proposed in this article to offshore wind turbines?
8. Is each layer in the distributed architecture indispensable? Can each layer of the architecture work independently?
Comments on the Quality of English LanguageSome grammar, punctuation, and word case issues in the paper need to be corrected. For example, there are some problems on line 52, line 96, and line 386. Please check them all.
Reviewer 3 Report
Comments and Suggestions for Authors
The main problem with this paper is that, although the results may be quite useful, the overall objective is unclear.
Already the title is mideading: The paper is not about PM in Manufacturing Systems" at all. It is about PM of assets in operation. Of course, any asset in use must have been manufactired before. However, the challenge of using the DT approach for PM of assets in operation, would be to apply the DT aproach of assets along their whole ife cycle, from the design, engineering, production down to the operational phase or even until recycling or re-manufacturing. There are modern approaches for this based upon Industrie 4.0 Asset Administration Shell (AAS, cf. ISO IEC 63278-x) and the Digital Product Passport (DPP) as a sub-model of the AAS.
There are open source implementations of the AAS concept to be considered in a DT framework.
The introduction does not show the purpose of the paper. It comes with the literature review which is strange.
Round 2
Reviewer 1 Report
Comments and Suggestions for Authors
No further questions
Reviewer 2 Report
Comments and Suggestions for Authors
This manuscript has made significant improvements compared to the previous version, and all corresponding issues have been addressed.
Reviewer 3 Report
Comments and Suggestions for Authors
thank you for having considered my comments